# Exploiting the Freshwater Shrimp *Neocaridina denticulata* as Aquatic Invertebrate Model to Evaluate Nontargeted Pesticide Induced Toxicity by Investigating Physiologic and Biochemical Parameters

**DOI:** 10.3390/antiox10030391

**Published:** 2021-03-05

**Authors:** Petrus Siregar, Michael Edbert Suryanto, Kelvin H.-C. Chen, Jong-Chin Huang, Hong-Ming Chen, Kevin Adi Kurnia, Fiorency Santoso, Akhlaq Hussain, Bui Thi Ngoc Hieu, Ferry Saputra, Gilbert Audira, Marri Jmelou M. Roldan, Rey Arturo Fernandez, Allan Patrick G. Macabeo, Hong-Thih Lai, Chung-Der Hsiao

**Affiliations:** 1Department of Chemistry, Chung Yuan Christian University, Chung-Li 320314, Taiwan; g10765013@cycu.edu.tw (P.S.); hieubtn90@gmail.com (B.T.N.H.); g10701304@cycu.edu.tw (G.A.); 2Department of Bioscience Technology, Chung Yuan Christian University, Chung-Li 320314, Taiwan; g10865014@cycu.edu.tw (M.E.S.); g10865016@cycu.edu.tw (K.A.K.); g10766011@cycu.edu.tw (F.S.); g10765017@cycu.edu.tw (A.H.); g10865013@cycu.edu.tw (F.S.); 3Department of Applied Chemistry, National Pingtung University, Pingtung 900391, Taiwan; kelvin@mail.nptu.edu.tw (K.H.-C.C.); hjc@mail.nptu.edu.tw (J.-C.H.); 4Department of Aquatic Biosciences, National Chiayi University, 300 University Rd., Chiayi 60004, Taiwan; wwwtt333@gmail.com; 5Faculty of Pharmacy and The Graduate School, University of Santo Tomas, Manila 1008, Philippines; mmroldan@ust.edu.ph; 6Laboratory for Organic Reactivity, Discovery and Synthesis (LORDS), Research Center for the Natural and Applied Sciences, University of Santo Tomas, Manila 1008, Philippines; reyarturo.tapia.fernandez@gmail.com; 7Center for Nanotechnology, Chung Yuan Christian University, Chung-Li 320314, Taiwan

**Keywords:** imidacloprid, insecticide, *Neocaridina denticulata*, shrimp, locomotion activity, cardiac activity, ecotoxicity

## Abstract

As a nicotinoid neurotoxic insecticide, imidacloprid (IMI) works by disrupting nerve transmission via nicotinic acetylcholine receptor (nAChR). Although IMI is specifically targeting insects, nontarget animals such as the freshwater shrimp, *Neocaridina denticulata*, could also be affected, thus causing adverse effects on the aquatic environment. To investigate IMI toxicity on nontarget organisms like *N. denticulata*, their physiology (locomotor activity, heartbeat, and gill ventilation) and biochemical factors (oxidative stress, energy metabolism) after IMI exposure were examined. IMI exposure at various concentrations (0.03125, 0.0625, 0.125, 0.25, 0.5, and 1 ppm) to shrimp after 24, 48, 72 h led to dramatic reduction of locomotor activity even at low concentrations. Meanwhile, IMI exposure after 92 h caused reduced heartbeat and gill ventilation at high concentrations. Biochemical assays were performed to investigate oxidative stress and energy metabolism. Interestingly, locomotion immobilization and cardiac activity were rescued after acetylcholine administration. Through molecular docking, IMI demonstrated high binding affinity to nAChR. Thus, locomotor activity and heartbeat in shrimp after IMI exposure may be caused by nAChR blockade and not alterations caused by oxidative stress and energy metabolism. To summarize, *N. denticulata* serves as an excellent and sensitive aquatic invertebrate model to conduct pesticide toxicity assays that encompass physiologic and biochemical examinations.

## 1. Introduction

Pesticides are a class of heterogeneous chemicals with significant public health benefits by increasing food production and decreasing food-borne and vector-borne diseases. However, studies have shown their adverse effects on nontarget organisms, including humans depending on the agent and the exposure [1]. Epidemiological studies suggest that exposure to pesticides may increase the incidence of respiratory diseases [2], neurologic dysfunctions [3], carcinogenicity [4], and reproductive disorders [5]. More than 20,000 pesticide products with nearly 900 active ingredients are registered as insecticides, miticides, herbicides, rodenticides, nematicides, and fungicides. Insecticides are considered the most useful pesticides among chemical or biological agents that control diseases caused by insect pests.

Imidacloprid [1-(6-chloro-3-pyridylmethyl)-2-nitroiminoimidazolidine] (IMI) belongs to a class of synthetic insecticides called neonicotinoids and is used widely to control insect pests on crops and fleas on domestic animals [6]. Contemporary insecticide products, like neonicotinoids, are often detected in aquatic systems [7]. Neonicotinoids potentially damage biological systems and antioxidant defenses. Exposure of susceptible organisms may lead to overstimulation in the nervous system, impairment of physiology, and death [8]. IMI is taken up by insects through contact and/or ingestion and binds to the nicotinic acetylcholine receptor (nAChR), allowing disruption of nerve impulses. It is selective to insect nAChR receptors and has much less pronounced effects in mammals [9,10]. The receptor anchors are ligand-gated ion channels and are involved in synaptic transmissions in the central nervous system [11]. There is evidence that IMI interaction with the α-subunit of nAChRs contributes to its partial agonist actions and its selectivity for insect nAChRs [12]. Sales of neonicotinoids are becoming fast-growing globally, brought about by low selectivity to insects and apparent safety for mammals [13,14]. Since IMI is a known substitute to other commercial pesticides, relative risks, and benefits associated with it must be compared to other pesticides.

Crustaceans such as the freshwater shrimp *Neocaridina denticulata* are used as animal models for screening toxicity because they exhibit rapid and sensitive responses to toxic substances. Additional characteristics of *N. denticulata*, include having small organismal size (2–3 cm), short reproduction period (4–5 months), and high larval production rate (50–100 individuals/breed) as beneficial aquatic indicators in assessing environmental pollution. *N. denticulata* has been used to determine pollutant effects, such as chlordane and lindane, on growth and reproductive hormones [15]. In 2005, the Environmental Protection Administration of Taiwan indicated its use during the conduct of acute toxicity assays to evaluate the safety levels of bodies of water or effluents coming from industrial districts [16].

Nontarget aquatic organisms can be exposed to IMI from spray drift, leaching, or run-off in agricultural contexts. Upon entry of IMI into aquatic environments, its dissipation may be aggravated by photolysis. While the possibility of neonicotinoid pesticide exposure exists in both targeted and nontargeted animals, there is limited data on studies illustrating IMI toxicity or adverse effects on nontarget aquatic organisms, particularly efforts describing effects on behavioral and biochemical alterations. Thus, we examined IMI’s possible acute effects on complex behavioral activity and neurotoxicity using biochemical, molecular biology, coupled with computational techniques such as molecular docking in the shrimp, *N. denticulata*, as an aquatic animal model.

It is noteworthy to mention that IMI acts by binding to nicotinic acetylcholine receptors and consequently disrupts nervous system functions [17]. It binds to nAChR in insect nervous tissues via neurotransmission dysregulation at cholinergic synapsis, thus causing death [18,19]. IMI selectively binds to nAChR [20] and prevents acetylcholine from binding to nAChR, causing blockage of regulated receptor binding functions of nAChR [21,22]. Acetylcholine is a neurotransmitter in the nervous system [23], particularly consumed in the neuromuscular junction and motor neurons to activate the muscles [23,24]. As a working hypothesis, we speculated that IMI might block nAChR binding of acetylcholine by selectively acting on the nicotinic acetylcholine receptors. In this study, we also illustrated that IMI affects both shrimp locomotion and cardiac activities. We also investigated the effects of exposing *N. denticulata* to acetylcholine, a biochemical that binds to nAChR, to check rescuing properties after IMI treatment. We thought that acetylcholine might increase locomotion and cardiac activity by saturating nAChR before binding to IMI.

## 2. Materials and Methods

### 2.1. Overview of Experimental Design to Conduct Acute Toxicity Test in Shrimp

Lack of specificity usually becomes the main problem of insecticides, thus, affecting both invertebrates and vertebrates other than insects in general [25]. The freshwater shrimp *N. denticulata* was used as a model in this study to represent aquatic invertebrates affected by insecticide carried away by overflow from terrestrial to receiving aquatic environments. As such, shrimp have been used as a model and bioindicator in assessing chemical contaminations in the environment due to high sensitivity [26].

The acute toxicity test in shrimp was conducted in six concentrations (0.03125, 0.0625, 0.125, 0.25, 0.5, and 1 ppm) of IMI from low to high concentrations, and a control. These six concentrations were chosen based on LC_50_ and EC_50_ data based on a previous procedure reported by our laboratory (see Section 2.3 and Section 2.4, respectively). Twelve shrimp were exposed in each group during the first experiment. The toxicity test (locomotor activity and heartbeat measurement) results were monitored at 24, 48, 72, and 96 h exposure to IMI. After 24 h of exposure to IMI, the treated shrimp’s locomotor activity was recorded using the ZebraBox instrument (Viewpoint Co., Lyon France). The locomotor activity of treated shrimp was observed every day until 72 h (day 3) post-exposure (Figure 1, red arrows) was reached. In addition to locomotor activity, heartbeat and gill ventilation (based on the maxilliped movement) were also measured to validate endpoints of the most sensitive markers for reporting IMI toxicity. The shrimp’s heart rate and gill ventilation were examined during 96 h exposure to IMI (Figure 1, blue arrows). After which, the remaining shrimp were frozen and sacrificed for biochemical testing using enzyme-linked immunoassay (ELISA) screens (Figure 1, green arrow). The EC_50_ dose of IMI was assessed after 96 h of IMI exposure (Figure 1). Complete immobilization occurred in shrimp after receiving higher concentrations (0.5 and 1 ppm) of IMI treatment. The EC_50_ was estimated at 0.51 ppm for IMI on inhibiting locomotor activity. Furthermore, to check whether acetylcholine could rescue IMI adverse effects in shrimp, a similar experimental design was used along with the addition of acetylcholine into treated shrimp. After IMI treatment for 24 h, treated shrimp were moved from IMI water, cleansed, and exposed to acetylcholine. Rescued shrimp locomotor and heartbeat were observed every day until day 3 (Figure 1). The concentrations used for the rescue experiment were 0.1 and 1 ppm of IMI along with acetylcholine 0.1 and 1 ppm. The differences between the first experiment (IMI toxicity test) and the rescue experiment are the total n number and the addition of acetylcholine in the rescue experiment. In the IMI toxicity test, the total n number for locomotor activity and heartbeat measurement were 12 shrimp for each concentration group, which is treatment (0.03125, 0.0625, 0.0125, 0.25, 0.5, and 1 ppm) and for control. Meanwhile, for the rescue experiment the total n number is 24 shrimp for each group, which is the treatment group (0.5 and 1 ppm) and the control group.

### 2.2. Animal Housing

*N. denticulate* with an average total length of 1–1.5 cm for experimental assays were provided by the Freshwater Resource Center of National Chiayi University, Chiayi City, Taiwan. Shrimp were stocked for 7 days at 26.5 °C in a recirculating system water under a 10/14 dark/light cycle to confirm healthy conditions before the assay. Circulating water in the aquarium was filtered by reverse osmosis at pH 7.0–7.5. Shrimp were fed twice a day (morning and evening) for 7 days prior to the conduct of the assay. All procedures in this study were approved by the Animal Ethics Committee of the Chung Yuan Christian University (Approval ID 107030).

### 2.3. Imidacloprid Treatment

Imidacloprid was obtained commercially as a 28.8% solution (Great Victory Chemical Industry Co. Ltd., Yunlin County, Taiwan). Its LC_50_ was reported as 0.91 μg/L in aquatic invertebrate [10]. IMI toxicity assays were initiated by diluting 28.8% stock solutions with water to working concentrations of 0.03125, 0.0625, 0.125, 0.25, 0.5, and 1 ppm. Exposure studies were conducted in 4 days in plastic tanks containing 20 shrimp with a dimension of 12 cm (L) × 9 cm (W) × 5 cm (H) filled with 200 mL working solution for each treatment.

### 2.4. EC_50_ Measurement for Immobilization

Concentrations at 50% of maximal effect (EC_50_) test for locomotion impairment were determined using six different concentrations of IMI of 0.03125, 0.0625, 0.125, 0.25, 0.5, and 1 ppm. Shrimp were incubated with different IMI concentrations, and the corresponding locomotor activities were measured every day after 24, 48, 72, and 96 h of exposure. EC_50_ curves for locomotion impairment at 24, 48, 72, and 96 h were constructed based on the immobilization of the shrimp from each test group.

### 2.5. Locomotion Tracking and Quantification

After IMI incubation, individual shrimp were placed into wells of a 6-well transparent plate (one shrimp per well/per concentration); each well with ≈10 mL of 0.03125, 0.0625, 0.125, 0.25, 0.5, or 1 ppm IMI) in two replicates. For video recording, each plate was placed into Zebrabox instrument (Viewpoint Co., Lyon, France) attached with a remote recording device equipped with a camera and infrared light-emitting where light can be controlled. The ViewPoint system, a video tracking software (Viewpoint Co., France), was set in tracking mode to record individual shrimp activity. Shrimp were initially habituated for 20 min, followed by 20 min of recording in the light cycle condition. After which, the total distance swam for each shrimp was calculated. Swimming speed thresholds were set and used to define three different speed thresholds as (1) bursting (>5.0 cm/s), is a short, intermittent, and powerful bout of activity; (2) cruising (1.0 ≤ s ≤ 5.0 cm/s), which captures commonly measured shrimp speeds; (3) freezing (<1.0 cm/s) will display the minimal activity of shrimp. All results were binned in 1-min intervals, resulting in 20 data points.

### 2.6. Heartbeat and Gill Ventilation (Based on Maxilliped Movement) Measurement

Heartbeat and gill ventilation (based on maxilliped movement) measurements were performed based on our previously established ImageJ-based dynamic pixel change method [27]. The video was captured using a digital charge-coupled device (CCD) (SK2700HDMI-T2, Zgenebio, Taipei, Taiwan) mounted on a dissecting microscope with a frame rate setting at 60 frames per second (fps). In this experiment, 12 shrimp were exposed to IMI. Each treated shrimp was recorded using CCD camera. *N. denticulata* after IMI exposure were used to conduct heart rate and gill ventilation assay using ImageJ as a major platform. A region of interest (ROI) was selected using a circle tool, and the selected ROI was targeted on either whole heart or maxilliped regions. Stack Difference and Time Series Analyzer Plugin were used to detect dynamic pixel changes over time. Graphic of beat rhythm was then performed using Origin 9.1 software (Originlab Corporation, Northampton, MA, USA). Peak times were calculated using the Peak Analyzer function in Origin 9.1 and processed using Microsoft Excel (Microsoft 2016 version, Seattle, WA, USA) to obtain time interval and beats per minute (bpm) data. Beats per minute were obtained by multiplying the time interval (in seconds) between two peaks by a factor of 60.

### 2.7. ELISA for Measuring Biomarker Expression

After completion of heartbeat and gill ventilation measurements, shrimp were sacrificed from both control and treatment groups to conduct the biochemical assays. Shrimp whole-body was homogenized in ice in volumes of 50 (*v/w*) phosphate-buffered saline (PBS) at pH 7.2 using a bullet blender tissue homogenizer (Next Advance, Inc., Troy, NY, USA). The samples were then centrifuged at 12,000 rpm for 20 min at 4 °C. The supernatant was collected and stored at −80 °C for total protein estimation and ELISA tests. BCA Protein Assay Kit (23225, Thermo Fisher Scientific, Waltham, MA, USA) was used to measure total protein concentration. In quantifying biomarkers expression, six ELISA kits were used to measure reactive oxygen species (ROS) related to oxidative stress, lipid peroxidation markers of 4-hydroxynonenal (4-HNE), and thiobarbituric acid reactive substances (TBARS) to determine lipid peroxidation in tissue samples. Additionally, energy metabolism bioassays for measuring glucose, pyruvate, and lactate levels were carried out. All commercial ELISA kits were purchased from Zgenebio Inc. (Taipei, Taiwan), and tests were conducted based on the manufacturer’s instructions. Color absorbance was analyzed at 450 nm wavelength using a microplate reader (Multiskan GO, Thermo Fisher Scientific, Waltham, MA, USA).

### 2.8. In Silico Methods

#### 2.8.1. Protein Preparation

The protein crystal structure of the Acetylcholine Binding Protein (Ls-AChBP, PDB ID: 2ZJU) from the great pond snail (*Lymnaea stagnalis*) was obtained from the Protein Data Bank (https://www.rcsb.org/, accessed on 1 February 2021) and its 3D structure (Figure A1) was incorporated into the UCSF chimera platform [28]. It was prepared initially by removing all nonstandard residues. Chains A and B were left in preparation for the docking to represent a subunit interface. The Acetylcholine Binding Protein structure was minimized by the steepest descent method (100 steps-step size 0.02 Å) and conjugate gradient method (10 steps-step size 0.02 Å).

#### 2.8.2. Ligand Preparation

Imidacloprid and acetylcholine were used as ligands for the docking experiments. The structures were then converted from SMILES format to SYBYL mol2 file using Avogadro (version 1.2.0), an open-source molecular builder, and were added to the UCSF Chimera platform for docking [29].

#### 2.8.3. Molecular Docking

UCSF Chimera was used as the platform for molecular docking studies [29]. The Gasteiger charge method computed using Amber’s Antechamber module added the missing hydrogen atoms and appropriate charges in the docking preparation [30]. The docking procedure employed flexible ligands into flexible active site protocol, allowing the ligand’s translational and rotational walk within the grid box [31]. Virtual screening of the prepared library was performed following the Broyden–Fletcher–Goldfarb–Shanoo (BFGS) algorithm of AutoDock Vina (version 1.5.6) [32]. Finally, the binding affinity was determined using UCSF Chimera and was visualized and analyzed through Biovia Discovery Studios (version 4.1).

### 2.9. Acetylcholine Rescue Experiment

Based on our results, IMI exposure induced a sharp immobilization in shrimp. An experiment was conducted to rescue the locomotor activity reduction in shrimp by adding acetylcholine to the IMI pretreated shrimp. Four test groups with 24 shrimp each were prepared and kept in 500 mL water tanks. The first group served as a control without any treatment. The other three groups were treated with 1 ppm IMI for 24 h. After 24 h, two groups for rescue treatment were then transferred to acetylcholine solution, one in a low concentration (0.1 ppm) and the other in a high concentration (1 ppm), while the remaining group was transferred to normal water without acetylcholine treatment. The shrimp locomotor activity, heartbeat, and gill ventilation rate from each group were recorded at 0, 24, 48, and 72 h post-exposure. For video recording, individual shrimp were placed into wells of a 6-well transparent plate (one shrimp per well/per concentration) and recorded for 20 min in three replicates. UMATracker, an open-source quantitative analysis for animal tracking [33], was used to track the shrimp’s locomotion activity. The video was loaded to UMATracker, and by using a filter generator, the video background was computed to retrieve and distinguish animal images. Animal images were converted to grayscale and binarized by adjusting the threshold value. By thresholding the image binarization algorithm, we obtained the output result of the converted video frame. At the tracking step, we set the number of animals in the video frame and ran the tracking algorithm to detect each animal’s position. After successfully tracking based on whole video frames, the data was saved as csv files format. These data contain XY coordinates and animals’ position per frame, which were processed using Microsoft Excel. The total distance activity from all groups was measured and compared. For heartbeat and gill ventilation measurement, the process has been explained above.

### 2.10. Biostatistics

The experimental values between control and IMI treated groups were compared. For heartbeat and gill ventilation rate, the statistical tests were analyzed by a two-way ANOVA test continued with Dunnet’s multiple comparisons test as the follow-up test. Rescue and acetylcholine exposure experiment data were expressed as percentages, with control day 0 as the comparison. For locomotor activity, time chronology assay, the statistical tests were conducted through the two-way ANOVA with the Geisser–Greenhouse correction with either Dunnett’s multiple comparison test or the Kruskal–Wallis test with Dunn’s multiple comparisons test as a follow-up test. For locomotor activity, most data were expressed as mean with SEM. We did not use median because several behavioral data have 0 value of median. In addition, if we used mean +/− 95% CI, some error bars will be less than 0. Therefore, mean ± SEM was reported to avoid confusion. Statistical tests were performed using GraphPad Prism (https://www.graphpad.com/, accessed on 1 February 2021).

## 3. Results

### 3.1. Imidacloprid Exposure Impaired Locomotor Activity in Shrimp

The quantitative comparison of different doses of IMI toxicity on the shrimp locomotor activity is shown in Figure 2. The total distances were calculated on the first 24, 48, and 72 h prior to IMI exposure. Based on the results, significant reductions in total locomotor activity were observed in all IMI-treated shrimp. This alteration was visible immediately after 1-day IMI exposure, supported by a significant decrease in the shrimp total distance traveled observed during the assay, including the lowest concentration group (0.03125 ppm) (Figure 2A,D). Furthermore, the statistically significant gaps in the shrimp total distance traveled between the control, and the treatment groups became more pronounced due to increased exposure concentrations and time (Figure 2B,C,E,F). At high concentration (1 ppm), shrimp locomotor activity consistently exhibited the lowest value compared to the other treated groups, starting from day one to day three of the test. The locomotor activity compromised after 3-day IMI exposure also can be found in Appendix A. From locomotor activity trajectory (Figure 2G), it will be more straightforward to see the shrimp treated with IMI showed sharp locomotor activity decline compared to the control group. Even the lowest concentration (0.03125 ppm) already exhibited stagnant locomotor activity (Figure 2A–G). Additionally, the total distance traveled in the control group was relatively high during the first day, while it was slightly reduced in the following days, probably due to food fasting. Overall, the results showed that IMI could significantly reduce shrimp locomotor activity based on factors such as treatment exposure, time, and dose.

### 3.2. Imidacloprid Exposure Impaired Heartbeat and Gill Ventilation in Shrimp

The effects of IMI exposure to heartbeat and gill ventilation are shown in Figure 3. To the best of our knowledge, we demonstrate for the first time the effect of pesticides on heartbeat and gill ventilation in *N. denticulata.* The heart is a vital organ for cardiovascular circulation in shrimp. Cardiovascular dysfunction has been used to evaluate environmental toxicants as significant risk factors in crustaceans [34,35]. The maxilliped is one of the three pairs of appendages located immediately behind the maxillae of shrimp and plays multiple functions on feeding and facilitating oxygen uptake [36,37]. Thus, heartbeat and maxilliped movement were selected as two additional endpoints for pesticide toxicity evaluation. The regions of interest (ROIs) in the heart and maxilliped for *N. denticulata* were highlighted in red and blue colors, respectively (Figure 3A).

In this study, shrimp exposed to IMI were not fed for the entire duration of the experiment. After shrimp exposure to IMI, the heartbeat and maxilliped movements were reduced compared to the control shrimp. At 96 h post-exposure, heartbeat rates of 0.5 and 1 ppm IMI treatments were significantly decreased (Figure 3C), while heartbeat intervals of 0.5 and 1 ppm IMI were increased (Figure 3D) compared to the control. On the other hand, a reduction in maxilliped movement was also observed in shrimp exposed to 0.25, 0.5, and 1 ppm IMI (Figure 3E,F). Moreover, heartbeat and gill ventilation movement in free-feeding shrimp not exposed to IMI as control were also examined. Interestingly, shrimp with sufficient nutrition had a higher heartbeat and maxilliped movement when fasted groups were compared (Figure 3C–F). The heartbeat rates of shrimp with sufficient food, lack of nutrition, and 1 ppm of IMI exposure were 194 ± 21.13, 135.2 ± 5.5, 86.56 ± 15.62 bpm, respectively. On the other hand, the gill ventilation movement in shrimp with sufficient food, lack of nutrition, and 1 ppm of IMI exposure were 404.5 ± 52.34, 207.3 ± 23.81, 119.9 ± 22.29 bpm, respectively. This result suggests feed conditions should be carefully controlled with consistency levels for shrimp heartbeat and maxilliped movement experiments to minimize potential variations.

In addition, we also extracted heartbeat and gill ventilation chronological patterns using the ImageJ software (Figure 3B). The dynamic pixel intensity is higher when the heart is pumping, and a higher peak rhythm is formed, while lower peak rhythm forms are noted when there is heart relaxation. Heartbeat and gill ventilation were calculated by dividing 1 min (60 s) with a time interval. A faster heartbeat will form additional peaks that will lead to a shortened time interval between two successive peaks. A slower heartbeat will form a longer time interval. In Figure 3B (red panel), the peak loosens as the heartbeat slows in shrimp treated with IMI. Similar results were noted in gill ventilation movement (Figure 3B blue panel). The heartbeat and gill ventilation move for control and IMI exposed shrimp can be found in Appendix A, respectively.

Further evaluation of heartbeat and gill ventilation movement alteration were checked using the Poincaré plot. The heartbeat and maxilliped movement were observed in the following groups: no starvation (control), starvation, and starvation with 1 ppm (IMI treatment). The Poincaré plot is a scattergram generated by plotting each R-R interval towards the previous one, based on our previous methods [38]. R-R interval is the interval between successive beats defined by heartbeats and gill ventilation. An increased and extensive scatter region visualized high variability. With the Poincaré plot, the irregularity can be analyzed quantitatively based on standard deviations (sd1 and sd2). The sd1 and sd2 are two descriptors of R-R intervals in the Poincaré plot determined by fitting an ellipse to the plotted shape. Sd1 is the standard deviation perpendicular to the line-of-identity, while sd2 is the standard deviation along the line-of-identity [39,40]. A higher value of sd1 and sd2 indicates high variability changes in heart rate and gill ventilation. Variability of heart rate in starvation (sd1 = 0.0314; sd2 = 0.0243) (Figure 4B) and IMI treatment (sd1 = 0.1122; sd2 = 0.0471) (Figure 4C) were higher compared to the control group (sd1 = 0.0179; sd2 = 0.0158) (Figure 4A). Meanwhile, highest variability of gill ventilation was displayed in the control group (sd1 = 0.0906; sd2 = 0.0697) (Figure 4D), followed by the IMI treatment (sd1 = 0.0703; sd2 = 0.0547) (Figure 4F), and the starvation group with the least variability (sd1 = 0.0192; sd2 = 0.0136) (Figure 4E) for gill ventilation movement. This result revealed that IMI treatment could cause irregularity in *N. denticulata* heartbeats and gill ventilation.

### 3.3. Biomarker Expression in Shrimp after Imidacloprid Administration

A biochemical assay was performed to identify the possible mechanism of locomotion immobilization, cardiac and gill ventilation alterations. For this purpose, we analyzed several essential biomarkers related to oxidative stress of reactive oxygen species (ROS), 4-hydroxynonenal (4-HNE), and thiobarbituric acid reactive substances (TBARS) to measure the extent of DNA damage after IMI exposure to *N. denticulata*. Pesticide-induced oxidative stress is mediated by excessive ROS production [41]. A recent study on *Drosophila melanogaster* indicated chronic IMI exposure could elevate the ROS level and damage the fly’s vision [42]. However, in this acute toxicity test for *N. denticulata*, we did not find any significant ROS level alteration after the short-term exposure to IMI (Figure 5A). The ROS level was found slightly higher at a concentration of 0.5 ppm but did not reach a significant difference with the control and other treatment groups. Oxidative stress also induces lipid peroxidation in cellular membranes and forms 4-HNE and TBARS as by-products. 4-HNE and TBARS test was used to quantify lipid peroxidation caused by oxidative stress in the biological system [43,44]. Similar to ROS results, there are no significant differences in 4-HNE and TBARS levels in all groups (Figure 5B,C). Energy metabolism of glucose, pyruvate, and lactate was also measured. No significant results for biomarkers in shrimp related to oxidative stress and energy metabolism for all tested concentrations were observed (Figure 5). We suggest that locomotion, heartbeat, and gill ventilation hypoactivity may not be associated with oxidative stress or alteration in energy metabolism. Hypoactivity could be caused by neurotransmitter disruption related to IMI’s mechanism of action as an inhibitor of acetylcholinesterase. IMI binds to acetylcholine receptors preventing transmission of impulses, resulting in acetylcholine accumulation [45]. Therefore, we experimented to restore normal acetylcholine levels to further investigate whether acetylcholine rescue could restore locomotion, heartbeat, and gill ventilation hypoactivity in *N. denticulata* after treatment with IMI.

### 3.4. Locomotion Impairment Can Be Rescued by Acetylcholine Administration

Nicotinic acetylcholine receptors (nAChRs) are ligand-gated ion channels that mediate fast cholinergic synaptic transmission in insects and vertebrate nervous systems [45,46]. The high affinity between IMI and nAChR led us to question whether overexposure to exogenous acetylcholine can compete with IMI for rescue. Thus, shrimp were initially incubated with 1 ppm IMI for 24 h to immobilize their locomotor activity. The shrimp were then transferred to a new tank containing freshwater, 0.1 ppm acetylcholine, or 1 ppm acetylcholine. The locomotor activity was monitored at day 0 for IMI exposure and days 1, 2, and 3 after exposure to acetylcholine for rescue. As such, 1 ppm IMI was found to sharply induce locomotor immobilization by exhibiting approximately 5-fold decrement compared to the untreated control group (Figure 6A,B). This immobilization phenotype can sustain for at least 3 days even after the shrimp were transferred to clean water (Figure 6A). The total distance in the IMI group rescue with freshwater was significantly reduced in four consecutive days, with a *p*-value < 0.0001. Interestingly, the compromised locomotor activity triggered after IMI treatment can be rescued by supplying 1 ppm exogenous acetylcholine (Figure 6A). The rescue effect can be observed even after 3 days of acetylcholine exposure (Figure 6A,E,I). From days 0, 1, and 2 after the rescue, locomotion from two rescue treatments (acetylcholine 0.1 and 1 ppm) showed lower total distance traveled with significant difference (*p* value < 0.0001) compared to the control. On the other hand, an increase in locomotor activity was observed in both rescue treatment groups on day 3. The 0.1 ppm acetylcholine treatment showed significantly higher total distance traveled than the IMI group treated with freshwater (Figure 6A,E,I). However, locomotion activity from this group was significantly lower compared to the control group. The 1-ppm acetylcholine group on day 3 displayed higher total distance traveled and displayed no significant difference than the control (Figure 6A). These results indicate that higher acetylcholine concentration with an equal balance to IMI (1:1 concentration) can restore shrimp locomotion activity.

### 3.5. Heartbeat and Gill Ventilation Rescue by Acetylcholine Administration

Imidacloprid exposure on shrimp showed that it could affect the shrimp by reducing the heart and gill ventilation rate (Figure 3). We attempted to rescue the heart and gill ventilation rate of the exposed IMI-treated shrimp with acetylcholine from this result. The rescue experiment was started by exposing shrimp with IMI for 24 h, and the heartbeat and gill ventilation rate were determined. After a day of exposure, IMI was removed from water in the shrimp tank. Water was replaced with acetylcholine with concentrations of 0.1 ppm, 1 ppm, while the other contained clean water. Through days 1–3, the shrimp was exposed to acetylcholine and normal water after treatment with IMI. Based on our observations, IMI exposure to shrimp caused decreased heart and gill ventilation rates on day 0 compared to the control shrimp, which was not exposed to IMI (Figure 7A,B). Our rescue experiment results are consistent with a previous result illustrated in Figure 3C,E. These results proved that IMI could affect heart and gill ventilation rate in invertebrate organisms like *N. denticulata*. After a day of exposure to IMI, the shrimp were removed from the IMI solution environment and treated with acetylcholine to rescue heart and gill ventilation rates. The result on day 1 (Figure 7A) showed that exposure to acetylcholine 0.1 and 1 ppm increases heartbeat, especially 1 ppm acetylcholine, which showed a higher heartbeat percentage. The shrimp treated with clean water also showed an increment in heart rate percentage (Figure 7A). Meanwhile, gill ventilation rates were reduced even after acetylcholine exposure on day 1. All the groups treated with acetylcholine at test concentrations of 0.1 ppm, 1 ppm along with the clean water showed reduced gill ventilation rates on day 1 (Figure 7B). For heart rate measurements starting at day 1, the increment already has been noted, and finally, on day 3, the heart rate in all the groups is already under similar levels with the control groups. Based on this result, we conclude that removal of IMI and exposure to acetylcholine could rescue the shrimp’s heart rate (Figure 7A). Furthermore, acetylcholine seems to work differently on gill ventilation rates. While heart rate has been successfully increased, gill ventilation is decreased significantly on day 1. With 1-ppm acetylcholine exposure, gill ventilation was increased on day 2 and decreased on day 3 (Figure 7B). This indicates that acetylcholine could affect heartbeat while it works differently on gill ventilation rate by decreasing it. From this result, we want to prove our hypothesis by exposing the shrimp directly to acetylcholine and check the effect.

Acetylcholine exposures at 0.1 and 1 ppm concentrations seemed to have similar effects as we found in the rescue experiment. Shrimp treated with 0.1 and 1 ppm acetylcholine showed higher heart rates than a control shrimp. After exposure to acetylcholine starting from day 0, the heart rate level was already higher than the control group. After exposure in 2 days, both acetylcholine-treated groups showed a significantly higher heartbeat percentage than the control groups. On day 3 of acetylcholine exposure, the shrimp showed higher heart rates than on day 2 exposure (Figure 7C). From this experiment, we were able to prove that acetylcholine exposure to shrimp can boost heartbeat rate. Compared to gill ventilation, the results showed an opposite trend compared to the heartbeat experiment. It seemed that acetylcholine works differently in gill ventilation because both low and high acetylcholine concentration exposure showed a decreased level of the gill ventilation rate (Figure 7D).

### 3.6. Molecular Docking

In probing and visualizing imidacloprid and acetylcholine’s binding mechanisms to nicotinic acetylcholine receptors, molecular docking studies were carried out. Nicotinic acetylcholine receptors (nAChRs) are ligand-gated ion channels mediating excitatory cholinergic neurotransmission in vertebrates and invertebrates [45,46]. Each nAChR is a pentamer that assembles as five subunits. Acetylcholine binds to extracellular ligand-binding domains at the subunit interfaces, which is formed by loops A–F. Upon binding acetylcholine and agonists, the central cation-permeable ion channel opens transiently, propagating nerve impulses.

Due to the shortage of public information for shrimp nAChR crystal structure, acetylcholine binding protein (AChBP) from the snail *Lymnaea stagnalis* is considered a surrogate marker of the ligand-binding domain in nAChRs for loops A–F, which are highly conserved and was used for our in silico experiments [28]. The simulation results of molecular docking show that the binding energy of acetylcholine binding protein to IMI is −6.0 kcal/mol. In contrast, the binding energy of acetylcholine is −4.2 kcal/mol to the nAChR. The intermolecular interactions of the atoms <2.5 angstroms from IMI and acetylcholine can be visualized in the docking poses seen in Figure 8A,B, respectively. The relatively strong binding was mainly observed through conventional hydrogen bonds between the nitrogen of the pyridine moiety in IMI with Trp143 and amidine *N*-nitro with the phenolic functionality of Tyr192. Other significant interactions include a *pi*-sigma bond between the pyridine and hydroxyl alkyl of Thr144 and multiple *pi*-alkyl bonds of the methylpyridine with amino acid residues Arg104, Leu112, Tyr113, and Met114 (Figure 8A). On the other hand, acetylcholine interacted only via weak van der Waals with Trp143 of nAChR (Figure 8B).

## 4. Discussion

Our study highlighted freshwater shrimp *N. denticulata* as an excellent laboratory animal model for a specific practice, such as ecotoxicology. This species is a decapod (crustacean) that is widely available in the aquarium industry as pets, aquatic plant cleaners, and fish bait. They are well-known to rapidly spread and expand their population in freshwater ecosystems [47]. *N. denticulata* has much potential to become a crustacean ecotoxicological model. Their characteristics are very much applicable to a wide variety of purposes and meet laboratory study criteria. The adult’s body length can grow to around 2.85 cm, making them experimentally simple trackable animals. The tissue-specific toxicity of insecticide can be addressed by conducting experimental validation at either biochemical or molecular levels in isolated tissues. They have also easily grown species that can tolerate a wide range of pH (6.5–8.0) and temperature (22–25 °C, up to 30 °C). These animals can be easily maintained in small freshwater tanks at room temperature with simple filtration and aeration for survival and reproduction [48]. Female *N. denticulata* is also highly reproductive, producing 20–30 fertilized eggs in 30 days post-mating. This makes them readily available and accessible to culture in the laboratory. Additionally, the draft genome [48] and transcriptome [49,50,51,52] of *N. denticulata* have been decoded, making them an ideal model for functional genomic studies [53]. This study investigated *N. denticulata* as an ideal aquatic nontarget invertebrate model for pesticide toxicity analysis. Given this scenario, studies to explore IMI physiologic effects in shrimp *N. denticulata*, a crustacean widely distributed throughout Taiwan [54], represent a relevant effort.

Imidacloprid is a member of the neonicotinoid class of insecticides [55]. It acts on the postsynaptic nicotinic acetylcholine receptors (nAChRs) located in the central nervous system of insects [56]. It functions as an agonist to the five-subunit arranged ligand-gated ion channel causing a biphasic response (initial increase in spontaneous discharge frequency followed by a complete block to nerve propagation) [57]. Its usefulness as an insecticide comes from its ability to selectively bind to nAChRs of insects primarily due to the insect’s differential sensitivity and vertebrate nAChR subtypes. However, its widespread use sparked concerns over other nontarget organisms [58]. Previous studies prove that IMI has a selective affinity to nAChR in insects’ nervous tissues [18,19]. Other studies also prove the opposite result where nontarget organisms can also be affected by IMI. Several studies have proven IMI’s ability to interact with nAChRs of the mouse brain, thus causing toxicity to mice [20]. This experiment showed that the bioactivation of IMI contributes to its toxicity to mammals [20]. Several studies also demonstrated that IMI exposure affects various fish species. For example, IMI was shown to be toxic in Japanese medaka larvae. This insecticide could induce developmental retardation and severely hamper the fitness of the fish [59]. Another study showed that IMI exposure of *Prochilodus lineatus* leads to DNA damage and oxidative stress in different tissues [60]. Another experiment also showed that acute IMI toxicity could cause inflammation and oxidative stress in common carps [61]. These previous studies proved that IMI might have adverse and toxic effects on nontarget organisms. Due to low soil binding and high-water solubility of neonicotinoids, aquatic ecosystems are at high risk of contamination, mainly due to run-off events. It is important to be aware of the adverse effect of IMI in aquatic organisms such as shrimp. Simultaneous studies of cardiac, gill ventilation, and behavioral effects induced by IMI are not well explored in any aquatic toxicity model, despite its wide use in agriculture and structural similarities with several neurotoxic drugs including nicotine and chlorpyrifos.

In this study, among the three assayed endpoints, we noted that shrimp locomotion activity could become the most sensitive parameter for chemical toxicity evaluation in terms of heartbeat, gill ventilation, and locomotion. Starting from a low dose at the ppb level, the locomotor activity already significantly decreased (Figure 2). Meanwhile, other parameters such as heartbeat and gill ventilation displayed significant alterations only at high dose IMI treatment at ppm levels. This result is consistent with our previous findings in zebrafish larvae and *Daphnia* neonates [62], showing locomotor activity alteration as a sensitive marker to conduct toxicological screening. Our study also corroborates a previous study in brown shrimp (*Farfantepenaeus aztecus*), showing behavior changes noticeably after IMI exposure. After exposure to IMI, the affected brown shrimp could not swim normally, stopped moving, and sprawled at the bottom of the tank, but their legs moved involuntarily and died [63]. We also observed similar behavioral alterations in freshwater shrimp *N. denticulata* after IMI exposure showing that at the lowest concentration of IMI (0.03125 ppm), paralysis-like symptoms have already been triggered. Another experiment conducted in aquatic insect *Simulium vittatum* also strengthened our result, where IMI treated insects showed abnormal behavior and muscle controls [64]. Other experiments conducted in zebrafish also showed IMI exposure resulting in neurobehavioral defects [65]. Reduced locomotion activity may be caused by IMI interaction with nAChR, thus preventing acetylcholine from binding with nAChR and causing dysregulation of the nervous system [20,21,22]. Dysregulation can lead to decreased locomotion activity. Paralysis also can be a reason why there is reduced locomotion activity [63]. After exhibiting abnormal swimming behaviors, shrimp legs will stop moving and become paralyzed. This leads shrimp to hunger and death [63]. Some of our treated shrimp also showed identical behaviors, and finally, some of them died at the final part of the test. The inability of treated shrimp to move and find any food can cause the shrimp to die. However, IMI may directly affect shrimp locomotor, inability to swim or moving the legs, and paralysis that primarily caused death in shrimp treated with IMI.

Another test we did was the cardiac rhythm for the shrimp. We noted that shrimp heartbeat and gill ventilation movement showed a decline when either lack of food and nutrition or IMI exposure. Previous studies in *Mytilus edulis* showed starvation to reduce the frequency of heartbeat and the rate of oxygen uptakes. After feeding, the frequency of heartbeat increased gradually to active levels [66]. In addition to starvation, some factors can affect the heartbeat such as environmental factors like temperature. A previous study has also proven that temperature can also affect the heartbeat. In *Daphnia*, heart rates display a good positive relationship with ambient water temperature [67]. In *M. edulis*, a high temperature was observed to increase the heartbeat. However, it will be reduced again after reaching a certain degree, and a lower temperature will reduce the heartbeat [66]. Interestingly, we found a consistent environmental condition to keep shrimp as an important factor in maintaining data reproducibility. Meanwhile, in the IMI treated group, we found that even at the lowest concentration, it could reduce heartbeat until the same level as shrimp starvation (Figure 3C). The highest concentration of IMI lowered heartbeat compared to the control and starvation group (Figure 3C). IMI exposure is known to cause a depressed heart rate. Previous experiments demonstrated that IMI exposure to *D. magna* could affect heartbeat by reducing the heart rate of *D. magna* in all of the tested concentrations [67]. In another *D. magna* study, a similar result was noted when *D. magna* was exposed to neonicotinoid insecticides. IMI, as neonicotinoid, has similar mechanisms and could affect *D. magna* by reducing heart rates after 48 h of exposure [68]. Our study disclosed similar results with other studies where *N. denticulata* with IMI caused a reduction in heart rate and gill ventilation compared to the control (Figure 3C,E). Another evidence that can prove our result is a study investigating the effects of exposing zebrafish to IMI where reduced heartbeat, blood flow, and malformation were observed [69]. As an invertebrate animal model in this study, shrimp has also demonstrated similar results after incubation with IMI. Our study proved that IMI indeed could attack and affect nontarget organisms like freshwater shrimp.

The reduction in locomotion activity, heart rate, and gill ventilation might also be linked with energy metabolism and oxidative stress. Hence, we investigated lipid peroxidation biomarkers such as ROS, TBARS, and 4-HNE to check whether oxidative stress is responsible for locomotion immobilization in *N. denticulata* after IMI exposure. Evidence collected from biochemical assays did not favor our previous hypothesis since IMI acute treatment had no adverse effect on energy metabolism and oxidative stress (Figure 5). Therefore, we proposed that reduced locomotion, heart rate, and gill ventilation are not influenced by biomarker alterations but are primarily contributed by acetylcholine binding blockage to the nAChRs due to IMI interference. The study from in vivo results was confirmed with in silico molecular docking, which revealed a higher binding affinity of IMI towards nAChR than acetylcholine (Figure 8). This result supports our hypothesis that there may be an IMI mechanism underlying in affected shrimp locomotion activity by binding with nAChR, thus preventing acetylcholine from interacting with nAChR. Our study supports a conserved mechanism of action between *N. denticulata* and other insect counterparts, showing superior binding affinity of IMI with nAChR compared to acetylcholine, thus impairing locomotion, heart rate, and maxilliped movements.

Since the molecular docking result revealed that IMI exhibits a stronger binding affinity to nAChR than acetylcholine, our hypothesis of acetylcholine overdosing might lead to competitive binding to nAChR. Thus, a rescue experiment was performed. IMI may remain on the water surface for a long time due to high water solubility [70]. IMI cannot be hydrolyzed by ACh esterase, causing interruption to nAChR leading to long-lasting effects, such as loss of reflexes, muscular weakness, and paralysis [71]. Consequently, it alters invertebrate movements and leads to starvation and death. A previous study revealed that IMI toxicity to freshwater amphipod *Gammarus pulex* could recover relatively fast after transferring to clean water [72]. However, our study found that locomotor immobilization triggered by IMI exposure could not be easily restored after transferring to clean water (Figure 6A). The heartbeat and gill ventilation rescue experiment also showed similar results with the locomotion. Even after moving IMI treated shrimp to fresh, clean water, the biological effect is maintained even after 3 days (Figure 8A,B). However, when acetylcholine was administered at either 0.1 or 1 ppm, the nontarget animals can recover from IMI sublethal effects, even though acetylcholine treatment needs 2 days to recover from IMI exposure. Meanwhile, for heartbeat and gill ventilation rescue experiments, acetylcholine could rescue the heartbeat faster compared to locomotion (Figure 7A). The heartbeat rescue experiment is supported by other results where acetylcholine-treated shrimp showed higher heartrate compared to the control (Figure 7C). The rescue experiment showed that exogenous acetylcholine exposure could compete with IMI to nACRs and restore the adverse effects triggered by IMI. Our results are consistent with a prior study conducted in insects showing IMI can be dissociated from nAChR by acetylcholine and other nicotinic ligands epibatidine, α-bungarotoxin, and methyllycaconitine [73]. Therefore, our findings support IMI reversible high binding affinity to nAChR. The rescue effect of acetylcholine needs quite some time to be showcased. The reason is that IMI is relatively stable and only slowly hydrolyzes in neutral water. Photodegradation of IMI in natural sunlight is also weak, making IMI in normal water persist for a long time [74]. Locomotion showed a rescue effect during the last day, while the heartbeat rate showed a rescue effect on day 2. The persistence of IMI in water seems to be why they still can bind with nAChR thus blocking acetylcholine as a natural neurotransmitter (the mechanism was summarized in Figure 9).

Finally, we successfully demonstrated that IMI treatment for 24 h could lead to immobilization, reduce heart rate, decrease gill ventilation, and ultimately cause death in shrimp. This insecticide’s adverse effect is specific to its target and could pose harmful effects to nontarget organisms like the shrimp *N. denticulata*. Butcherine et al. reported that neonicotinoid exposure might raise adverse effects on wild shrimp fisheries and aquaculture productivity due to adverse effects on feeding and lipid contents in tiger shrimp [75]. Bardran et al. also reported that IMI exposure at corresponding EPA benchmark concentrations could sharply reduce survival rates and delay molting in juvenile brown shrimp (*Farfantepenaeus aztecus*) [63]. All these reports support our study that showcases IMI adverse effects on *N. denticulata* in locomotion and cardiac rhythm.

## 5. Conclusions

We successfully created and demonstrated the imidacloprid adverse effect to non-target organism such as freshwater shrimp (*N. denticulata*). Shrimp exposed to imidacloprid insecticide showed immobilization, reduce heart rate, decrease gill ventilation, and ultimately could cause death. This insecticide proven to be toxic not only to target organism, but also harmful to non-target organism such as freshwater shrimp. According to this experiment, the use of chemical such as insecticide that could harm environment should be regulated with caution. The use of imidacloprid in the future should pay attention to our environment. We also provide possible mechanism to explain imidacloprid could trigger immobilization with rescue experiment. In this consideration, our finding opens a new avenue to perform translational studies beneficial to shrimp fisheries and aquaculture industries. In situations where cultured shrimp with high economic value been polluted by IMI, the possibility of conducting acetylcholine rescue to prevent significant economic loss is useful and relevant.

## Figures and Tables

**Figure 1 antioxidants-10-00391-f001:**
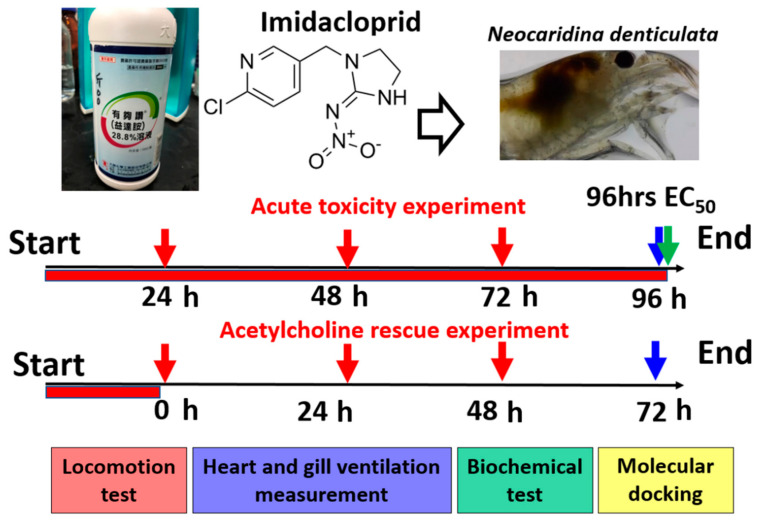
Overview of experimental design to conduct imidacloprid (IMI) acute toxicity and acetylcholine rescue test in shrimp (*Neocaridina denticulata*). For the acute toxicity test, IMI was exposed to *N. denticulata* for 96 h. By 24, 48, and 72 h post-exposure, the locomotory activity changes of IMI-treated shrimp were measured. After 96 h, the heartbeat and gill ventilation (measured by maxilliped beating rate) of IMI-treated shrimp were measured. The experimental animals were subjected to enzyme-linked immunosorbent assay (ELISA) assay for biomarker expression measurements. For the rescue experiment, shrimp were pretreated with 1 ppm IMI for 24 h and transferred to either clear water or acetylcholine solution. After 0, 24, 48, and 72 h post-exposure, the rescued shrimp’s locomotory activity changes were measured and compared.

**Figure 2 antioxidants-10-00391-f002:**
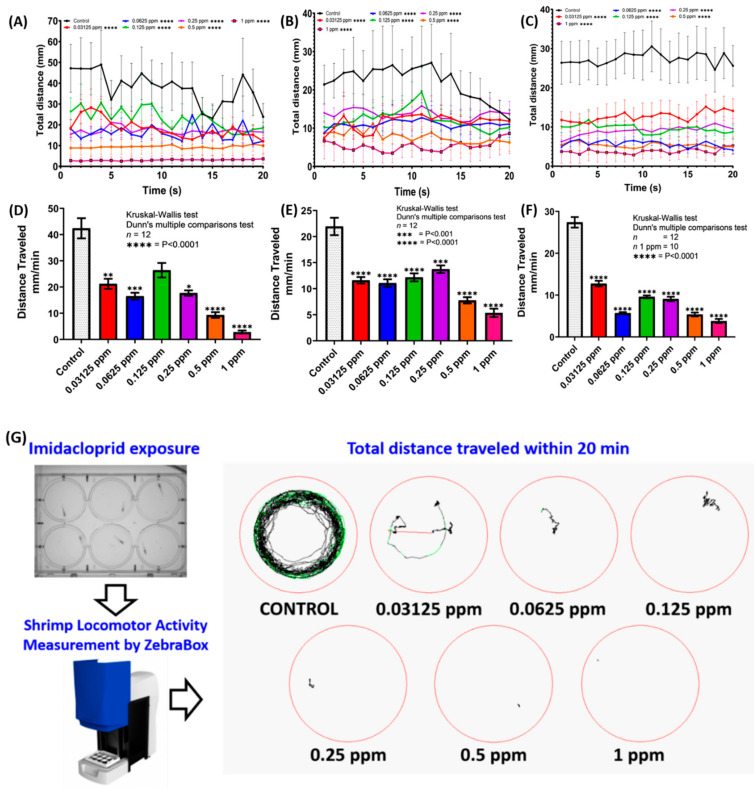
(**A**–**C**) Average distance traveled per minute by untreated and imidacloprid (IMI) treated shrimp (*Neocaridina denticulata*) after 24, 48, 72 h exposure, respectively, in given concentrations (0.03125 ppm (red), 0.0625 ppm (blue), 0.125 ppm (green), 0.25 ppm (purple), 0.5 ppm (orange), 1 ppm (pink)). The data are expressed as the mean with SEM and were analyzed by two-way ANOVA with the Geisser–Greenhouse correction. Dunnett’s multiple comparison test for comparing all treatments with control was carried out to observe the main column (IMI) effect. (**D**–**F**) Total average distance traveled per minute for 20 min locomotor activity test by the control (untreated), and IMI treated shrimp after 24, 48, 72 h exposure. (**G**) The trajectory of shrimp treated with imidacloprid within 20 min video recording by Zebrabox instrument. The data are expressed as mean with SEM and were analyzed by the Kruskal–Wallis test continued with Dunn’s multiple comparisons test as a follow-up test (*n* = 12 for all of the groups, except 1 ppm group on the 72 h exposure time interval (*n* = 10) (* *p* < 0.05; ** *p* < 0.01; *** *p* < 0.001; **** *p* < 0.0001).

**Figure 3 antioxidants-10-00391-f003:**
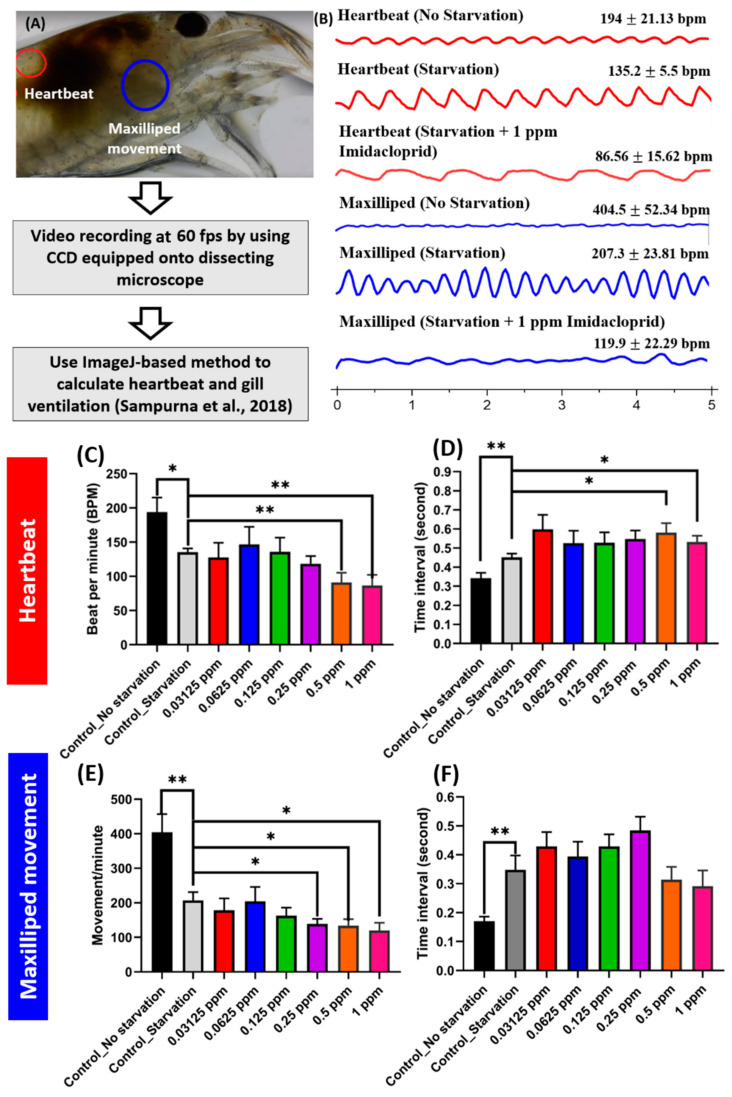
The heartbeat and gill ventilation (based on maxilliped beating) in shrimp (*Neocaridina denticulata*) are affected by starvation and pesticide pollution. (**A**) Overview of the method used to measure heartbeat and maxilliped beating in the shrimp (left panel). (**B**) The dynamic pixel changes over time for the heart (red color) or maxilliped (blue color) with no starvation, starvation, or imidacloprid (IMI) treatments. Evaluation of IMI toxicity at different doses by monitoring (**C**) heart rate, (**D**) heartbeat interval, (**E**) gill ventilation rate, and (**F**) gill ventilation interval in *N. denticulata*. The data are expressed as mean with SEM and were analyzed using two-way ANOVA along with Dunnet’s multiple comparisons as a follow-up test. (* *p* < 0.05; ** *p* < 0.01).

**Figure 4 antioxidants-10-00391-f004:**
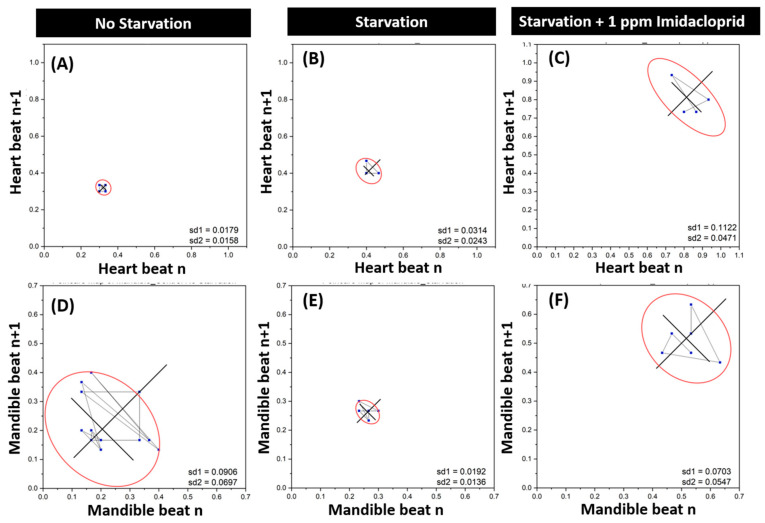
Evaluation of imidacloprid (IMI) toxicity at different doses by monitoring heartbeat and gill ventilation regularity. Poincaré plot showing the heartbeat (**A**–**C**) and maxilliped beating (**D**–**F**) regularity when treated with no starvation (**A**,**D**), starvation (**B**,**E**), or IMI (**C**,**F**) treatments. The data are expressed as mean with SEM, and the one-way ANOVA tested the significance with a nonparametric Mann–Whitney test.

**Figure 5 antioxidants-10-00391-f005:**
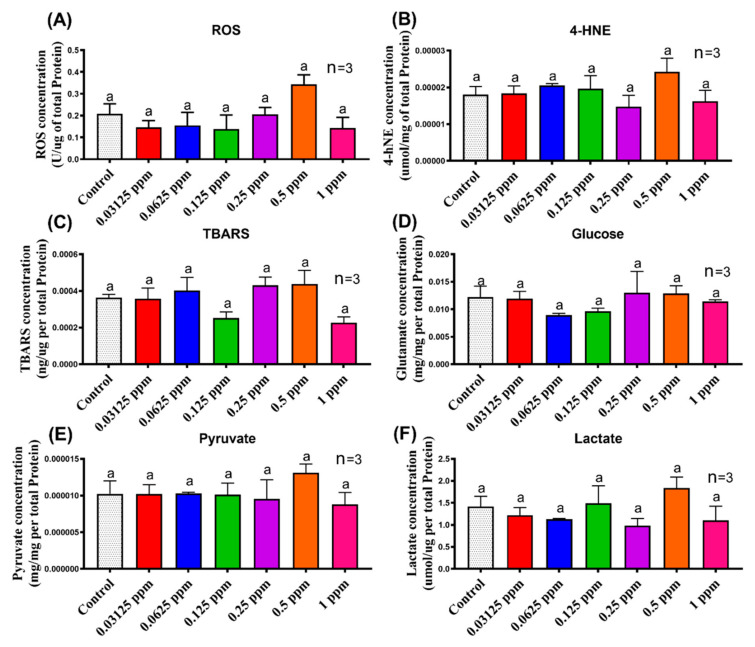
Evaluation of biomarker content in shrimp (*Neocaridina denticulata*) by enzyme-linked immunosorbent assay (ELISA) after exposure to different imidacloprid (IMI) concentrations. Here we used (**A**) ROS, (**B**) 4-HNE, and (**C**) TBARS to evaluate oxidative stress, (**D**) glucose, (**E**) pyruvate, and (**F**) lactate to evaluate energy metabolism. The data are expressed as mean with SEM. Statistical analysis was by one-way ANOVA test followed by Tukey post-hoc test (a is used as a label, different letter (a) above the error bars indicate a significant difference with *p* < 0.05).

**Figure 6 antioxidants-10-00391-f006:**
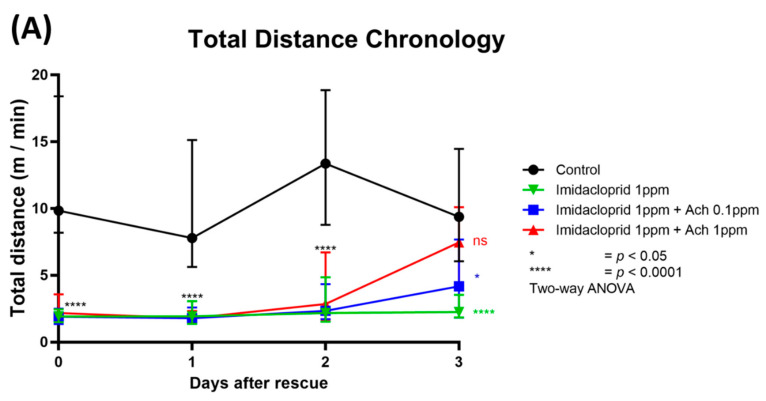
The rescue experiment showing acetylcholine can restore locomotor immobilization triggered by imidacloprid (IMI) in shrimp (*Neocaridina denticulata*). (**A**) Total distance chronology of shrimp treated with IMI and rescue using acetylcholine from days 0–3. (**B**) Total distance traveled chronology for 20 min locomotor activity test on day 0 of IMI treated shrimp. (**C**) Total distance traveled chronology of shrimp treated IMI rescue with acetylcholine for 20 min locomotor activity test on day 1. (**D**) Total distance traveled chronology of shrimp treated IMI rescue with acetylcholine for 20 min locomotor activity test on day 2. (**E**) Total distance traveled chronology of shrimp treated IMI rescue with acetylcholine for 20 min locomotor activity test on day 3. The data are expressed as mean with SEM and were analyzed using two-way ANOVA with Geisser–Greenhouse corrections. Dunnett’s multiple comparison test for comparing all treatments with control was carried out to observe the main column (IMI) effect. (**F**) Total distance traveled locomotor activity test on day 0 of IMI treated shrimp. (**G**) Total distance traveled by shrimp treated IMI rescue with acetylcholine locomotor activity test on day 1. (**H**) Total distance traveled by shrimp treated IMI rescue with acetylcholine locomotor activity test on day 2. (**I**) Total distance traveled by shrimp treated IMI rescue with acetylcholine locomotor activity test on day 3. The data are expressed as mean with SEM and were analyzed by Kruskal–Wallis test along with Dunn’s multiple comparisons test as follow-up test (*n* = 24 for all of the groups on day 0, except 1 ppm IMI group on the day 1, 2, and 3 exposure time intervals (*n* = 22)). (* *p <* 0.05; **** *p* < 0.0001).

**Figure 7 antioxidants-10-00391-f007:**
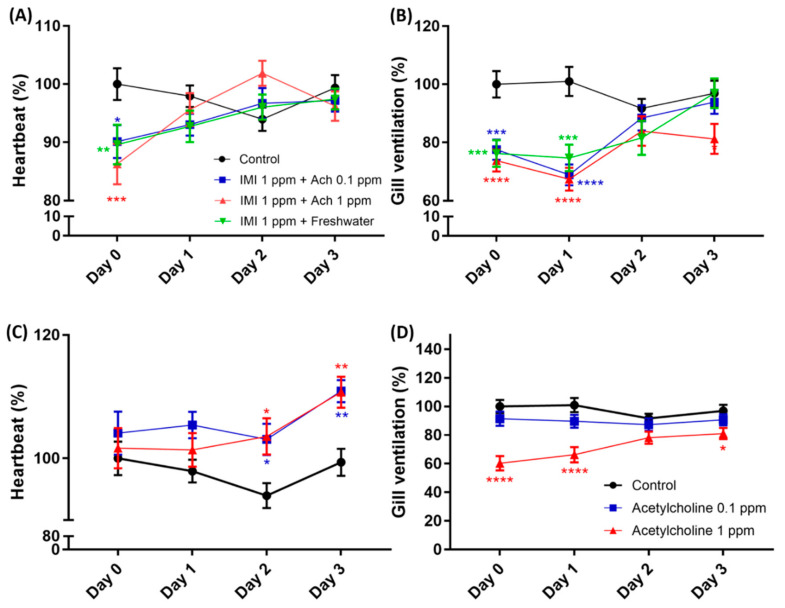
The rescue experiment showing acetylcholine can restore the heartbeat and gill ventilation beating in shrimp (*Neocaridina denticulata*) after being treated with imidacloprid (IMI). (**A**) The heart rate of shrimp after exposure to IMI for 24 h (day 0) and rescue by acetylcholine exposure through days 1–3. (**B**) Gill ventilation rate of shrimp after exposure to IMI for 24 h (day 0) and rescue by acetylcholine through days 1–3. The data are expressed as mean with SEM and were analyzed using two-way ANOVA along with Dunnet’s multiple comparisons as a follow-up test (*n* = 24 for the control group on day 0 through 3; *n* = 24 for 1 ppm IMI rescue with 0.1 ppm acetylcholine group on days 0–2, except for day 3 (*n* = 23); *n* = 24 for 1 ppm IMI rescue with 1 ppm acetylcholine group on day 0, on day 1–2 *n* = 23, while on day 3 *n* = 22; *n* = 22 for 1 ppm IMI rescue with fresh water on day 0, while from day 1 through day 3 *n* = 21). (**C**) Heart rate of shrimp after exposure to acetylcholine for 3 days (Control positive). (**D**) Gill ventilation rate of shrimp after exposure to IMI for 3 days (Control positive). The data are expressed as mean with SEM and were analyzed by two-way ANOVA along with Dunnet’s multiple comparisons as a follow-up test (*n* = 24 for the control group on day 0 through 3; *n* = 24 for 0.1 ppm acetylcholine on day 0–2, except for day 3 (*n* = 23); *n* = 24 for 1 ppm acetylcholine group on day 0, *n* = 23 on day 1, while *n* = 22 on days 2–3). (* *p* < 0.05; ** *p* < 0.01; *** *p* < 0.001; **** *p* < 0.0001).

**Figure 8 antioxidants-10-00391-f008:**
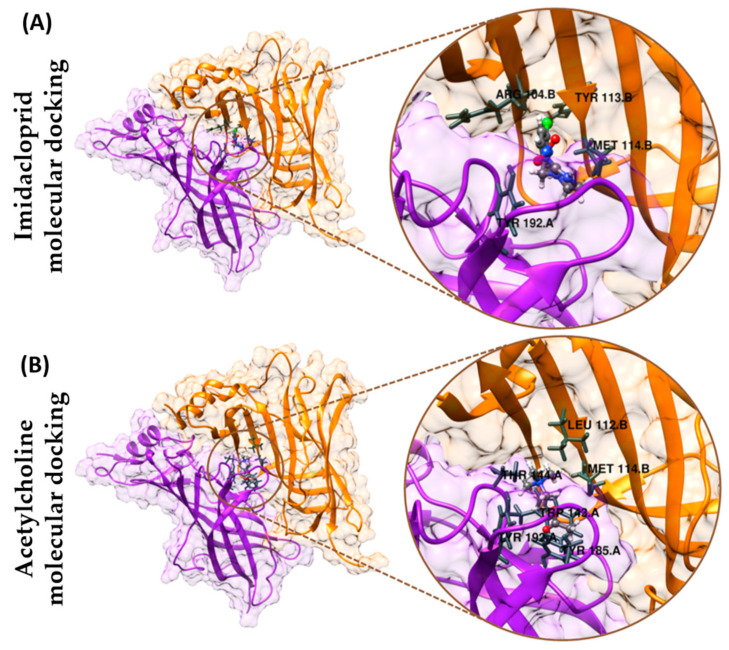
Molecular docking experiment to compare the binding affinity of imidacloprid (IMI) and acetylcholine to nicotinic acetylcholine receptor (nAChR). (**A**) The docking pose includes chains A (purple) and B (orange), representing two of the five subunits that make nAChR. The binding of IMI is seen in the interface of the two chains. IMI, at <2.5 angstroms, exhibited intermolecular interactions with Tyr192 of chain A and Arg104, Tyr113, Met114 of chain B. A wide-angle view (left) and a focused image (right) are illustrated. (**B**) The binding of acetylcholine is seen in the chain A (purple) and B (orange) interface. At <2.5 angstroms, acetylcholine exhibited intermolecular interactions with Trp143, Thr144, Tyr185, Tyr192 of chain A and Leu112, Met114 of chain B. A wide-angle view (left) and a focused image (right) are illustrated.

**Figure 9 antioxidants-10-00391-f009:**
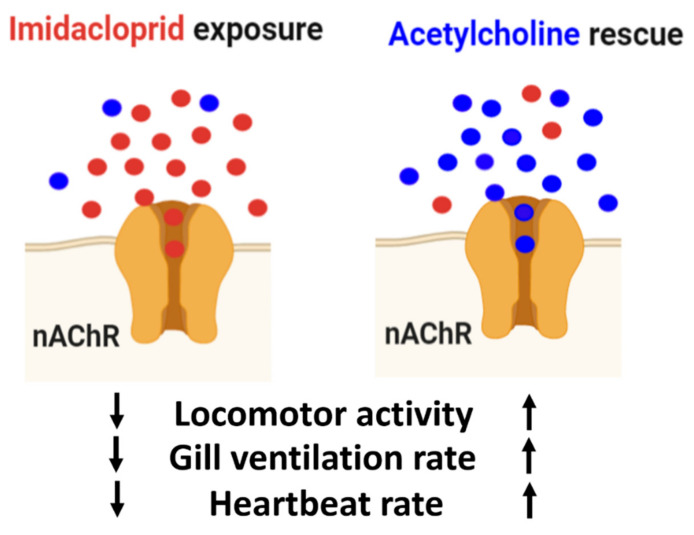
Model to explain why imidacloprid (highlighted by red color) can trigger locomotor compromise in shrimp and has been rescued by acetylcholine (highlighted by blue color) administration. Exogenous imidacloprid has high affinity to nicotinic acetylcholine receptor (nAChR) and compete with endogenous acetylcholine. The nAChR blockage by imidacloprid leads to signal transduction blockage and reduces locomotor activity, gill ventilation and heartbeat rates. However, when the shrimp has been exposed to excess amount to exogenous acetylcholine, those physiological compromises can be rescued.

## Data Availability

Original data and videos can be obtained from authors upon request.

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
