# Peer review of "Exploiting the Freshwater Shrimp Neocaridina denticulata as Aquatic Invertebrate Model to Evaluate Nontargeted Pesticide Induced Toxicity by Investigating Physiologic and Biochemical Parameters"

_antioxidants, 2021, doi:10.3390/antiox10030391_

Round 1
Reviewer 1 Report
This is an excellent study/manuscript. I have some comments/questions regarding the experimental design and statistical analyses. I also offer a few minor editorial comments.
METHODS
Experimental units (replications, n = ) ? -- In the "Overview of experimental design..." section (2.0, pp 3-4), which includes/references Figure 1, or in the Biostatistics section (2.9, p 7), the replication is not defined. Some of the descriptions of experiments or measurements are noted -- sections 2.2, Imidacloprid treatment ("tanks containing twenty shrimps"); 2.4, Locomotion tracking./quantification ("one shrimp per well"); and 2.8, Acetylcholine rescue experiment ("Four test groups with twenty-four shrimps each...") -- but it is not clearly stated whether the number of shrimp = n = number of replications.
Locomotion tracking and quantification (section 2.4, p 5) / ANOVA Design Question? – The first sentence of this section reads, “After IMI incubation, individual shrimp were placed into wells of a 6-well transparent plate (one shrimp per well (per concentration); each well with ~10 ml of 0.03125, 168 0.0625, 0.125, 0.25, 0.5, or 1 ppm IMI) in three replicates.”. Am I correct in thinking that the three replicates (n = 3) are three 6-well plates? If so, then the experimental design would be a randomized complete block ANOVA (a type of mixed-model ANOVA: treatment = fixed effect; plate/block = random effect) – not a two-way ANOVA. And the blocks (plates) would function as replications. If there were not three plates/blocks used in the experiment, then what are the treatment replications?
Biostatistics section (2.9, p7) / Median +/- 95% CI vs Mean +/- SEM ? – (1) For some analyses, the median/95%CI is used; for others, the mean/SE is used. I’m assuming that the median was used for sample data that were non-normal and/or heteroscedastic, but your reason for using two different statistics should be stated in the Methods. (2) To be consistent, why not use the 95% CI for both the mean and median (or the SE for the mean or median)?
Biostatistics -- Two-Way ANOVA with Dunnett’s test or Kruskal-Wallis test with Dunn’s test? ANOVA and Dunnett’s MCP test are parametric tests, which is appropriate for data/groups that exhibit normal distributions and homogeneous variances. Kruskal-Wallis and Dunn’s MCP test are nonparametric tests, which may be used for data with non-normal distributions. State your rationale for using ANOVA/Dunnett’s for some analyses and Kruskal-Wallis/Dunn’s for others.
RESULTS
Figure 2 / Reporting the median for ANOVA/Dunnett’s test results? The median is typically used to summarize data analyzed with nonparametric tests (e.g., Kruskal Wallis test). Why not use the mean for ANOVA results?
Figure 2 / Reporting Mean for Kruskal-Wallis/Dunn’s test results? The mean is an appropriate statistic for summarizing data analyzed with parametric tests (e.g., ANOVA). Why not use the median for Kruskal-Wallis test results?
EDITORIAL COMMENTS
Line 107 / Text: “Lack of specificity usually becomes the main problem of insecticides; thus, affecting … / Note: Replace semicolon with comma.
Lines 141-142 / Text: “After 0, 24, 48, and 72 hr post-exposure…” / Revised --> After 0-, 24-, 48-, and 72-hr post-exposure…
Note: Same type of revision needed in other parts of manuscript
Line 157 / Text: “…twenty shrimps…” / Revised -->…20 shrimps… / Note: Write numerals for numbers >10 (unless it’s the first word of a sentence). This occurs in other parts of the paper.
Line 114 / Text: “… and control.” / Revised --> and a control.
Line 208 / Text “… carriedout.” / Revised --> … carried out
Lines 215-216 / Text “The protein crystal structure information of the great pond snail (Lymnaea stagnalis) Acetylcholine Binding Protein (Ls-AChBP, PDB ID: 2ZJU) was obtained from …” / Revised -> The crystal structure of the Acetylcholine Binding Protein (Ls-AChBP, PDB ID: 2ZJU) in the great pond snail (Lymnaea stagnalis) was obtained from …
Line 554 / Text “…rapidly widespread and expand…” / Revised --> …rapidly spread and expand
Author Response
Comments and Suggestions for Authors
This is an excellent study/manuscript. I have some comments/questions regarding the experimental design and statistical analyses. I also offer a few minor editorial comments.
METHODS
Experimental units (replications, n = ) ? -- In the "Overview of experimental design..." section (2.0, pp 3-4), which includes/references Figure 1, or in the Biostatistics section (2.9, p 7), the replication is not defined. Some of the descriptions of experiments or measurements are noted -- sections 2.2, Imidacloprid treatment ("tanks containing twenty shrimps"); 2.4, Locomotion tracking./quantification ("one shrimp per well"); and 2.8, Acetylcholine rescue experiment ("Four test groups with twenty-four shrimps each...") -- but it is not clearly stated whether the number of shrimp = n = number of replications.
The author already changed the method section according to reviewer’s comment in this revised version. Thank you for your carefully reviewing and constructive comments to increase the quality of the paper (line 129-135).
Locomotion tracking and quantification (section 2.4, p 5) / ANOVA Design Question? – The first sentence of this section reads, “After IMI incubation, individual shrimp were placed into wells of a 6-well transparent plate (one shrimp per well (per concentration); each well with ~10 ml of 0.03125, 168 0.0625, 0.125, 0.25, 0.5, or 1 ppm IMI) in three replicates.”. Am I correct in thinking that the three replicates (n = 3) are three 6-well plates? If so, then the experimental design would be a randomized complete block ANOVA (a type of mixed-model ANOVA: treatment = fixed effect; plate/block = random effect) – not a two-way ANOVA. And the blocks (plates) would function as replications. If there were not three plates/blocks used in the experiment, then what are the treatment replications?
Thank you for comments. In the first locomotor activity test (Fig. 2), we used 12 n number in 6-well plate with one shrimp per well. Therefore, we did duplicate in the first locomotor activity. Meanwhile, for the rescue experiment on locomotor activity, we used 24 n number, which includes three replicates and we used the same 6-well plate in order to reduce randomized effects. In addition, the author also checked the chronology which is the time effect in locomotor activity. In this case there were two factors in the locomotor test, concentration and time effect, with this reasoning the author chose two-way ANOVA to analyze the data in the locomotor activity test. We already revised the method section and added more details to prevent any confusion (line 267-278).
Biostatistics section (2.9, p7) / Median +/- 95% CI vs Mean +/- SEM ? – (1) For some analyses, the median/95%CI is used; for others, the mean/SE is used. I’m assuming that the median was used for sample data that were non-normal and/or heteroscedastic, but your reason for using two different statistics should be stated in the Methods. (2) To be consistent, why not use the 95% CI for both the mean and median (or the SE for the mean or median)?
Thank you for the comment to improve the quality of the manuscript. The author realized that there are some typographical errors reflected in the information indicated in the figure legend. We already used same statistics such as Mean +/- SEM. We did not use median because several behavioral data have 0 value of median. In addition, if we used Mean +/- 95% CI, some error bars will be less than 0. Therefore, we prefer to express the data as mean ± SEM to avoid confusions. We will correct these and revise the biostatistics section accordingly (line 267-278).
Biostatistics -- Two-Way ANOVA with Dunnett’s test or Kruskal-Wallis test with Dunn’s test? ANOVA and Dunnett’s MCP test are parametric tests, which is appropriate for data/groups that exhibit normal distributions and homogeneous variances. Kruskal-Wallis and Dunn’s MCP test are nonparametric tests, which may be used for data with non-normal distributions. State your rationale for using ANOVA/Dunnett’s for some analyses and Kruskal-Wallis/Dunn’s for others.
Thank you for this comment. We really appreciate this input to increase the quality of the manuscript. The author used Kruskal –Wallis/Dunn’s test for data with non-normal distribution, which was already done in this experiment. Meanwhile, for locomotor activity, especially in chronology data, the authors used Two-Way ANOVA with Dunnett’s test because in the locomotor activity test even though the data are not normally distributed, we reported two factors which is concentration and time effects. However, the author is also fully aware that the Two-Way ANOVA is a parametric test, therefore to obtain more reliable data the authors applied Geisser-Greenhouse corrections since the sphericity (equal variability of differences) was not assumed. The author already included the rationale in the method section in this revised version (line 267-278).
RESULTS
Figure 2 / Reporting the median for ANOVA/Dunnett’s test results? The median is typically used to summarize data analyzed with nonparametric tests (e.g., Kruskal Wallis test). Why not use the mean for ANOVA results?
Thank you for your comment in order to improve the quality of the manuscript. This case is similar with previous question according to biostatistics, the author will renew and added more information in biostatistics section to prevent any confusion (line 302-313).
Figure 2 / Reporting Mean for Kruskal-Wallis/Dunn’s test results? The mean is an appropriate statistic for summarizing data analyzed with parametric tests (e.g., ANOVA). Why not use the median for Kruskal-Wallis test results?
Thank you for your comment in order to improve the quality of the manuscript. This case is similar to the previous comment related to biostatistical treatments, the authors have addressed the comment accordingly and added more information in the biostatistics section to avoid confusion (line 302-313).
Reviewer 2 Report
Congratulations!
It was a pleasure to review the paper.
Author Response
Thanks for your reviewing and suggestions
Reviewer 3 Report
The manuscript is devoted to an interesting and relevant topic and can be in demand by a wide range of readers. The research topic is fully consistent with the journal topic. The article has a clear purpose, logical structure, and practical significance. The research methods are modern and adequate. The results are presented sequentially, well-illustrated, and explained.
However, I suggest introducing some adjustments to improve the quality of the paper.
1) The abstract should be a total of about 200 words maximum. (301 words)
2) Some examples of typos, mistakes:
Line 26. for pesticidal toxicity – for pesticide toxicity
Line 130-131. for rescue experiment – for the rescue experiment (this phrase is used in each of the last three sentences of the paragraph)
Lines 131-132. acetylcholine 01. And 131 1 ppm.
Line 151. A period is missing at the end of the sentence.
Line 178. will displays – will display
Line 208. carriedout – carried out
Pay attention to the grammar of lines 319-323.
Line 359. variability. . With – Two periods at the end of a sentence.
3) Perhaps it is necessary to write why such dosages were chosen. Lines 113-114. The acute toxicity test in shrimp was conducted in six concentrations (0.03125; 0.0625; 0.125; 0.25; 0.5; and 1 ppm) of IMI from low to high concentrations, and control.
Author Response
Comments and Suggestions for Authors
The manuscript is devoted to an interesting and relevant topic and can be in demand by a wide range of readers. The research topic is fully consistent with the journal topic. The article has a clear purpose, logical structure, and practical significance. The research methods are modern and adequate. The results are presented sequentially, well-illustrated, and explained.
However, I suggest introducing some adjustments to improve the quality of the paper.
- The abstract should be a total of about 200 words maximum. (301 words)
Thank you for the suggestion, the author already made adjustments on the abstract section as per reviewer’s comments.
2) Some examples of typos, mistakes:
Line 26. for pesticidal toxicity – for pesticide toxicity, Line 130-131. for rescue experiment – for the rescue experiment (this phrase is used in each of the last three sentences of the paragraph), Lines 131-132. acetylcholine 01. And 131 1 ppm., Line 151. A period is missing at the end of the sentence., Line 178. will displays – will display, Line 208. carriedout – carried out, Pay attention to the grammar of lines 319-323, Line 359. variability. . With – Two periods at the end of a sentence.
Thank you for your carefully reviewing and detailed inputs to make our manuscript better. We already made changes according to reviewer’s comments.
3) Perhaps it is necessary to write why such dosages were chosen. Lines 113-114. The acute toxicity test in shrimp was conducted in six concentrations (0.03125; 0.0625; 0.125; 0.25; 0.5; and 1 ppm) of IMI from low to high concentrations, and control.
Thank you for your suggestion. We are pleased to read your comments and suggestions. The authors are aware that it is important to include more explanation regarding the chosen concentration used in the experiments. The author already included this information suggested by the reviewer in this revised version (line 106-109).